# An Application of Artificial Neural Network for Predicting Threshing Performance in a Flexible Threshing Device

**Lan Ma** [1,2,3] , **Fangping Xie** [1,3,*], **Dawei Liu** [1,3], **Xiushan Wang** [1,3] and **Zhanfeng Zhang** [4]

1 College of Mechanical and Electrical Engineering, Hunan Agricultural University, Changsha 410102, China; malan@caas.cn (L.M.); liudawei@hunau.edu.cn (D.L.); hnwxs@hunau.edu.cn (X.W.)
2 Institute of Bast Fiber Crops, Chinese Academy of Agricultural Sciences, Changsha 410205, China
3 Hunan Key Laboratory of Intelligent Agricultural Machinery Corporation, Changsha 410102, China
4 Changsha Zichen Technology Development Co., Ltd., Changsha 410221, China
* Correspondence: hunanxie2002@163.com

**Abstract:** Rice is a widely cultivated food crop worldwide, and threshing is one of the most important operations of combine harvesters in grain production. It is a complex, nonlinear, multi-parameter physical process. The flexible threshing device has unique advantages in reducing the grain damage rate and has already been one of the major concerns in engineering design. Using the measured test database of the flexible threshing test bench, the rotation speed of the threshing cylinder (RS), threshing clearance of the concave sieve (TC), separation clearance of the concave sieve (SC), and feeding quantity (FQ) are used as the input layer. In contrast, the crushing rate ($Y_P$), impurity rate of the threshed material ($Y_Z$), and loss rate ($Y_S$) are used in the output layer. A 4-5-3-3 artificial neural network (ANN) model, with a backpropagation learning algorithm, was developed to predict the threshing performance of the flexible threshing device. Next, we explored the degree to which the inputs affect the outputs. The results showed that the R of the threshing performance model validation set in the hidden layer reached 0.980, and the root mean square error (*RMSE*) and the average absolute error (*MAE*) were less than 0.139 and 0.153, respectively. The built neural network model predicted the performance of the flexible threshing device, and the regression determination coefficient $R^2$ between the prediction data and the experimental data was 0.953. The results showed revealed that the data combined with the ANN method is an effective approach for predicting the threshing performance of the flexible threshing device in rice. Moreover, the sensitivity analysis showed that RS, TC, and SC were crucial factors influencing the performance of the flexible threshing device, with an average relative importance of 15.00%, 14.89%, and 14.32%, respectively. FQ had the least effect on threshing performance, with an average threshing relative importance of 11.65%. Our findings can be leveraged to optimize the threshing performance of future flexible threshing devices.

**Keywords:** rice; flexible threshing cylinder; artificial neural network; threshing clearance of concave sieve; separating clearance of concave sieve; feeding quantity; threshing performance

## 1. Introduction

Rice is one of the four main staple food crops in China, with a perennial planting area of 30 million hectares [1]. Mechanized rice production relies heavily on the harvest process as an essential step. Threshing is a key link in the rice harvesting process; it is a complex, nonlinear, and uncertain process, with several influencing parameters and large nonlinearity [2,3]. The impact of threshing on rice determines how much grain is lost during the harvest and processing stages. Double cropping rice in southern China has a short harvesting duration. The performance parameters of the threshing and separation device directly affect the operation quality of the combined rice harvester, i.e., the core working component. The longitudinal axial threshing device is characterized by long threshing time, smooth threshing process, good adaptability, and relatively soft threshing

effect, and it is broadly used in combined harvesters [4]. Researchers in agricultural mechanization are interested in the flexible threshing tooth due to its lower impact force and rate of damage to the cracked grains compared to its rigid counterpart [5]. For this reason, it is suitable for increasing the synthesis benefit in grain production [6]. Several scholars have studied the application of flexible materials in agricultural engineering. In 1972, Duane L et al. [7] designed a self-made collision test device to analyze the effects of corn grain velocity, collision surface material, collision angle, and other parameters on the extent of grain collision damage. One study found that when the impact surface was polyurethane, the damage degree of the grain was one-fifth of that when the impact surface was steel, and one-sixth of that when the impact surface was concrete. This is an inaugural study focusing on the effect of flexible materials on grain, demonstrating the benefit of flexible materials in reducing grain damage degree. Shi Qingxiang et al. [8] performed a comparative study on the flexible and rigid threshing elements, demonstrating that flexible threshing with flexible teeth made of flexible materials can extend the threshing time and reduce grain breakage with feasible flexible threshing. Xie Fangping et al. [9] utilized polyurethane plastic cylindrical strips as the teeth of flexible threshing rods to conduct a dynamic analysis of the threshing of flexible rod teeth. Consequently, they found that the indexes of flexible threshing, for instance, non-removal rate and impurity rate, were similar to those of rigid rod teeth threshing, and the crushing rate was significantly lower than that of rigid rod teeth threshing. Ren Xuguang et al. [10] analyzed the threshing process of rice using the conservation law of capacity and noted that it is conducive to rice threshing when the flexible teeth periodically hit the ear of rice, and a resonance response occurred. Su Yuan et al. [11] modified the conventional Q235 carbon steel teeth into nitrile rubber composite nail teeth and polyurethane rubber nail teeth. The test found better grain removal performance of nitrile rubber composite nail teeth than that of polyurethane rubber nail teeth and traditional carbon steel nail teeth. Geng Duanyang et al. [12] designed a cross-axial flow flexible corn threshing device. To realize flexible and low-damage threshing of corn ears, the threshing element combined a structure of flexible nail teeth and an elastic short grain rod. Li Yibo et al. [13] performed a bench test to explore the effect of composite nail teeth of different outer materials on the threshing performance and self-wear resistance of corn ear. The results showed that the rubber composite nail teeth had the best comprehensive effects in threshing and self-anti-fraying performance, the breakage rate of maize was lower compared with that of traditional carbon steel nail teeth, and the non-threshing rate of maize was similar to that of traditional carbon steel nail teeth, thus meeting the conditions of technical specifications for threshing quality evaluation of maize harvester. Fu Jun et al. [14] established a rigid–flexible coupled wheat threshing arch tooth. Under similar operating conditions, the damage rate of the rigid-flexible coupled arch tooth was significantly reduced, unlike that of the standard arch tooth, with significant loss reduction and threshing effect. Qian Zhenjie et al. [15] introduced the increase and decrease constraint strategy to establish a multi-friction dynamic model of flexible threshing teeth on grains. As a consequence, it was observed that the continuous normal striking force and repeated minor tangential kneading force of flexible teeth on grains combined to reduce the grain damage rate. Reports on the longitudinal axial flow threshing cylinder with a hollow core and flexible rod teeth used in rice threshing are limited. Flexible threshing can reduce the crushing rate of rice grains and, thus, developing a comprehensive and accurate evaluation model of flexible threshing has important theoretical value and practical significance.

In recent years, the artificial neural network (ANN) has achieved desired performance and high accuracy in predicting laboratory data because of its capacity to describe non-linear systems. As a result, it is widely applied in the fields of mathematics, engineering, medicine, economy, environment, and agriculture [16], particularly where some traditional modeling methods have failed [17]. Artificial neural network technology has been utilized in harvester systems by some researchers [18,19]. Nonetheless, few studies have been conducted on the threshing performance of a flexible threshing device using artificial neural networks. Due to the uncertainty of the threshing condition and the complexity of the

factors affecting the threshing device, the threshing performance prediction is a nonlinear problem affected by multiple factors. Nevertheless, the BP neural network is a nonlinear dynamic system [20,21] with powerful nonlinear [22] and generalization capacity and can identify complex relationships among the data [23]. Herein, parameters [24] affecting the performance of the threshing device and threshing performance indicators [25,26] were based on the parameters reported by several studies.

In the laboratory-based flexible threshing bench test, the rotated speed of the threshing cylinder, threshing clearance of the concave sieve, and separating clearance of the concave, as well as feeding quantity, were selected as the inputs of the model based on the BP neural network. The neural network model was established between inputs and their threshing characteristic of the crushing rate, impurity rate of threshed material, and entrainment loss rate. Further, the threshing performance index was predicted under different parameters. The objectives of this study included: (1) Determining the feasibility of artificial neural network technology in predicting the threshing performance of the flexible threshing device and providing executable procedures for an artificial neural network model for practical application; (2) Investigating the effect of artificial neural network geometry and some internal parameters on model performance; (3) Exploring the relative significance of factors influencing threshing performance through sensitivity analysis.

## 2. Materials and Methods

### 2.1. Test Materials and Equipment

The plots with basically similar crop growth rates were selected as the experimental sampling area. The rice variety tested was Xiangzaoxian 24. Table 1 shows the main material characteristics of the rice. The rice flexible threshing test was conducted in the Agricultural Machinery Engineering Training Center of Hunan Agricultural University from July 11 to 18, 2022. Figure 1 shows the test equipment, and Table 2 shows the equipment parameters.

**Table 1.** Main physical characteristic parameters of harvesting rice.

| Rice Varieties | Plant Height/mm | Panicle Length/mm | Middle Stem Diameter/mm | Middle Stem Wall Thickness/mm | Number of Shoots per Ear | Number of Grains per Ear | Thousand-Grain Mass/g | Stem Moisture Content/% | Grain Moisture Content /% | Yield per Unit Area /kg·hm⁻² | Ratio of Grass to Grain |
|---|---|---|---|---|---|---|---|---|---|---|---|
| Xiangzaoxian No. 24 | 833 ± 64 | 182 ± 12 | 32.18 ± 0.3 | 0.4 ± 0.1 | 12 ± 1.6 | 110 ± 22.9 | 30.02 ± 1.0 | 55.68 ± 4.8 | 22.42 ± 0.8 | 6230 | 1:(0.83 ± 0.1) |

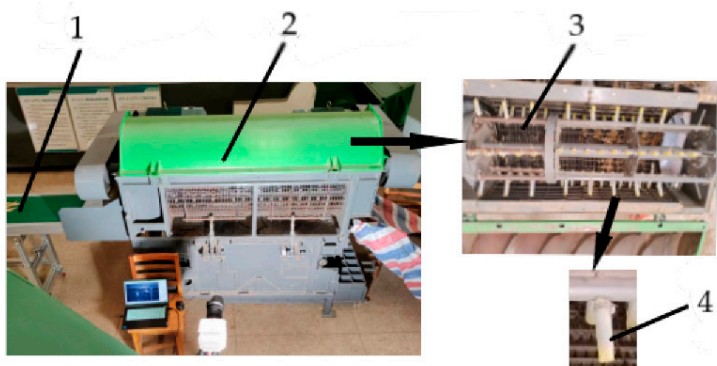

**Figure 1.** Flexible threshing experiment device. 1. Feeding device 2. Threshing device 3. Threshing cylinder 4. Flexible threshing teeth.

**Table 2.** Parameter table of flexible threshing device.

| Parameters | Values |
| --- | --- |
| The total length of the cylinder/mm | 1935 |
| Threshing cylinder diameter/mm | 620 |
| Cylinder speed/(r·min$^{-1}$) | 400–1500 |
| Threshing clearance of concave sieve/mm | 0–60 |
| Separating clearance of concave Sieve/mm | 0–60 |
| Feeding rate/(kg·s$^{-1}$) | 0.5–5 |

### 2.2. Test Method

The test was conducted following GB/T 5262—2008 and GB/T 5982—2005.

The thousand-grain quality was determined using the national standard method to explore grain and stem water content and according to the GB 5519-85 "grain, oil test thousand grain weight determination method".

The plots with similar crop growth were selected as the sample areas. Rice plants were artificially fed uniformly into the longitudinal axial flow threshing drum. In the multi-factor experiment, the material of each group weighed 10 kg. Three parallel tests were performed using similar parameter combinations, and the average value was taken. The performance evaluation indexes of the system were categorized into the crushing rate, impurity rate of threshed material (impurity rate for short), and entrainment loss rate. The mix that was threshed was collected in the receiving box located under the adaptable threshing mechanism. The mix released from the end of the cylinder was accumulated with the help of a tarpaulin attached to it. After each parallel test, the crushing rate and impurity rate of the threshing system were calculated using the mix, which was discharged into the receiving box. The mixture discharged onto the tarpaulin attached to the end of the cylinder was analyzed to determine the entrainment loss rate. The calculation formulas of the crushing, impurity, and entrainment loss rates are, respectively:

$$Y_P = \frac{W_P}{W_X} \times 100\% \tag{1}$$

$$Y_Z = \frac{W_{XZ}}{W_{Xh}} \times 100\% \tag{2}$$

$$Y_S = \frac{W_W}{W} \times 100\% \tag{3}$$

where $Y_P$ is the crushing rate, %; $W_P$ is the mass of crushed grains in the sample, g; $W_X$ represents the total grain weight in the sample, g; $Y_Z$ is the impurity rate of threshed material, %; $W_{XZ}$ is the impurity mass in the extruded sample, g; $W_{Xh}$ is the total mass of extruded samples, g; $Y_S$ is the entrainment loss rate, %; $W_W$ is the grain mass discharged from the tail of the drum, g; $W$ is the grain weight of each group of test extracts, g.

### 2.3. Building the ANN Model

2.3.1. Development of Neural Network Model

One of the most commonly used neural network models is the BP neural network, which utilizes the BP algorithm. Even the most complex nonlinear relationship completely approximates it. The information is dispersed and stored in the neurons of the network. The computation is extremely fast due to parallel processing. Since neural networks are self-learning and adaptive, they can deal with uncertain or unknown systems. This system is excellent when simultaneously processing both quantitative and qualitative information. It can coordinate a wide range of input information relations and is, thus, ideal for fusion and multimedia applications. A well-trained artificial neural network

can function as a predictive model for a specific application, which is a data processing system inspired by biological neural systems. The predictive power of an ANN is derived from training on experimental data, which is then validated using independent data. Artificial neural networks can relearn and adapt to improve their performance by updating data availability [27]. The structure and operation of ANNs have been described by numerous authors [28]. The modeling used in feedforward neural networks for prediction was designed to capture the correlation between the historical model inputs and their corresponding outputs. This is accomplished by repeatedly feeding the model examples of input/output relationships and adjusting the model coefficients (i.e., connection weights) to minimize the error function between the historical output and the model-predicted outputs.

This article follows the procedure of the artificial neural network model as described by Maier and Dandy [29]. They include determining model inputs and outputs, dividing and preprocessing available data, selecting an appropriate network architecture, optimizing connection weights (training), setting stopping criteria, and validating the model. A typical algorithm flow diagram is shown in Figure 2. In this work, all calculations and programming were executed in MATLAB (R2016a, 9.0.0.341360). The data used to calibrate and validate the neural network model were obtained from the bench field measurements of the flexible threshing experiment device and the corresponding information on the feeding amount and material characteristics. The data cover a wide range of variation in different operating parameters types and threshing properties. The database comprises a total of 25 individual cases. The statistics of the input and output parameters used for the artificial neural networks are shown in Table 3. Figure 3 is a database of all the threshing performance metrics for the ANN.

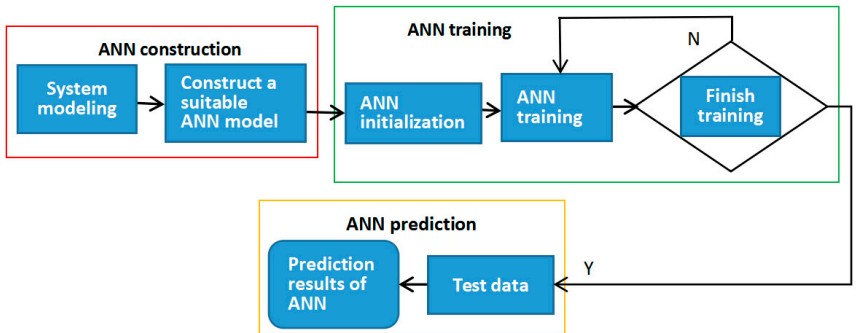

**Figure 2.** The artificial neural network algorithm flow chart.

**Table 3.** Statistical criteria of the input parameters and performance attributes (output parameters) used in the ANN model.

| Parameters | | Statistical Criteria | | | | | |
|---|---|---|---|---|---|---|---|
| | | Minimum | Maximum | Average | Standard Deviation | Median | Variance |
| Training set | inputs | 1 | 800 | 190.4500 | 300.8486 | 25 | $9.0154 \times 10^4$ |
| | outputs | 0.0490 | 1.1960 | 0.4142 | 0.4074 | 0.2070 | 0.1660 |
| Validation set | inputs | 2 | 800 | 183.3750 | 291.1664 | 30 | $8.4778 \times 10^4$ |
| | outputs | 0.0490 | 1.1960 | 0.3984 | 0.4231 | 0.1920 | 0.1790 |
| Testing set | inputs | 1.5 | 800 | 197.7750 | 311.6720 | 35 | $9.7139 \times 10^4$ |
| | outputs | 0.04093 | 0.9970 | 0.3995 | 0.4093 | 0.1920 | 0.1675 |

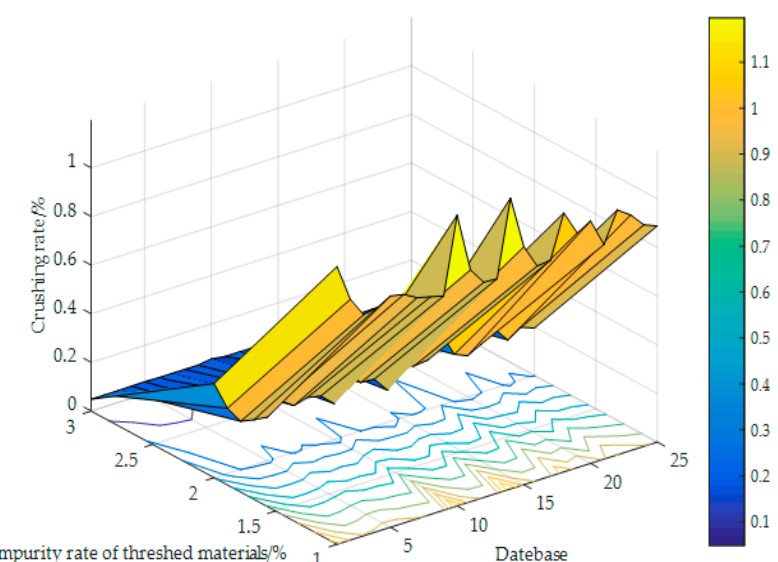

**Figure 3.** The threshing performance index database for the artificial neural network.

2.3.2. Model Inputs and Outputs

A thorough comprehension of the determinants of threshing performance is required to obtain accurate threshing performance prediction. The rotational speed of the cylinder is closely associated with the performance of the thresher because high-speed results in cracking of the grain, and low-speed leads to unthreshed grain. Moreover, the threshing clearance and separating clearance of the concave sieve significantly influence the threshing performance. Furthermore, the size of the feeding quantity is closely correlated to the threshing characteristics [30].

The primary factors affecting threshing performance include the rotational speed of the cylinder, threshing clearance of the concave sieve, separating clearance of the concave sieve, and feeding quantity. Other factors include total threshing power consumption and grain moisture content that contribute to a lesser degree, thus considered secondary. Grain moisture content was excluded in this work since the tests were conducted under specific moisture content conditions during the harvest period.

The aforementioned factors, i.e., the rotational speed of cylinder (RS), threshing clearance of concave sieve (TC), separating clearance of concave sieve (SC), and feeding quantity (FQ), were introduced to the ANN as the model input variables. On the other hand, the crushing rate ($Y_P$), impurity rate of threshed materials ($Y_Z$), and entrainment loss rate ($Y_S$) were the output variables. Sensitivity analysis was conducted on the trained network to identify the input variables with the most significant impact on threshing performance predictions.

Sensitivity analysis (Figure 4) was based on the validation set, where the input variable was RS, TC, SC, and FQ. To normalize the input variables, the value of the input variable was first changed, the trained network was introduced, the maximum and minimum output values were recorded, the difference between the maximum and the minimum value was computed, the difference to the maximum value was calculated, before finally taking the mean of all ratios as the sensitivity of the classification variable. Lastly, the sensitivity size was compared to establish the sensitivity of each categorical variable to the output variable. The sensitivity analysis results will be discussed later.

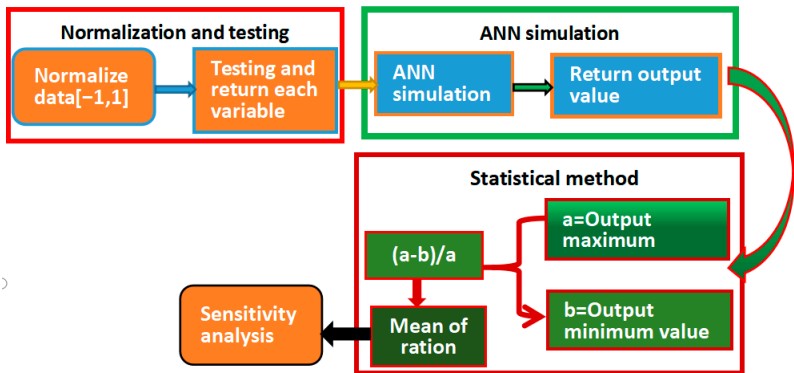

**Figure 4.** The sensitivity analysis flow.

### 2.3.3. Data Division and Preprocessing

The database was randomly divided into three sets, i.e., training, testing, and validation. A training set was used to construct the neural network model, whereas an independent validation set was used to estimate model performance in the deployed environment [31]. In total, 60% of the data were used for training, 20% for testing, and 20% for validation. Table 4 shows the orthogonal test for different levels as well as the data ranges used for the ANN model variables.

**Table 4.** Orthogonal test factor level table used for Artificial Neural Network Model Variables.

| Factors<br><br>Levels | Rotational Speed of Cylinder, RS (r/min) | Threshing Clearance of Concave Sieve, TC (mm) | Separating Clearance of Concave Sieve, SC (mm) | Feeding Quantity, FQ (kg/s) |
|---|---|---|---|---|
| 1 | 600 | 15 | 15 | 1.0 |
| 2 | 650 | 20 | 25 | 1.5 |
| 3 | 700 | 25 | 35 | 2.0 |
| 4 | 750 | 30 | 45 | 2.5 |
| 5 | 800 | 35 | 55 | 3.0 |

Notably, it is critical to preprocess the data into an appropriate format before applying it to the ANN. Preprocessing the data by scaling is crucial in ensuring that all variables receive equal attention during training. The output variables must be scaled to commensurate with the limits of the transfer functions used in the output layer. Although scaling the input variables is not necessary, it is often recommended [32]. Here, the input and output variables were scaled between $-1.0$ and $1.0$, as the purelin sigmoidal transfer function was used in the output layer.

### 2.3.4. Model Architecture

Determining the network architecture is one of the crucial and challenging tasks in the development of ANN models because it requires the selection of several hidden layers and the number of nodes in each of these.

The number of model inputs and outputs restricts the number of nodes in the input and output layers. The input layer of the ANN model developed in this work had four nodes, one for each of the model inputs (i.e., a rotational speed of cylinder (RS), threshing clearance of concave sieve (TC), separating clearance of concave sieve (SC), feeding quantity (FQ)). On the other hand, the output layer had three nodes (i.e., crushing rate ($Y_P$), impurity rate of threshed materials ($Y_Z$), and entrainment loss rate ($Y_S$)) representing the measured value of threshing performance.

Figure 5 shows the basic elements of an artificial neuron. Artificial neurons mainly comprise weight bias and activation functions. The BP neural network is the most popular and widely used artificial neural network architecture [33]. It involves an input layer, one

or more hidden layers, and an output layer. Evidence suggests that a network with a threshold, at least one S-shaped hidden layer, and a linear input layer can approximate any rational number [34]. Mathematical expressions and interpretations of artificial neural networks can be referred to in reference [35].

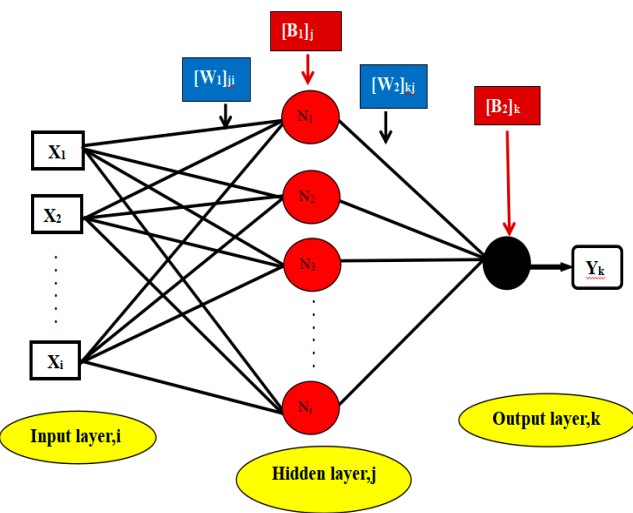

**Figure 5.** Schematic diagram of an artificial neural network.

The activation function introduces nonlinearity into the neural network, making it more powerful than the linear transformation. The Levenberg–Marquardt algorithm is the most commonly used multi-layer perception training algorithm. It is a gradient descent technique [36] used to reduce the error of specific training patterns. The network was built using the Levenberg–Marquardt backpropagation technique. Tansig is a common nonlinear activation function for nodes in the hidden layer. Figure 6 depicts the architecture of the artificial neural network system described in this paper. $W$ is a weight matrix for the hidden and output layers, and $N_{ij}$ is a node that computes a weighted sum of its inputs and passes the sum through a soft nonlinearity or activity function.

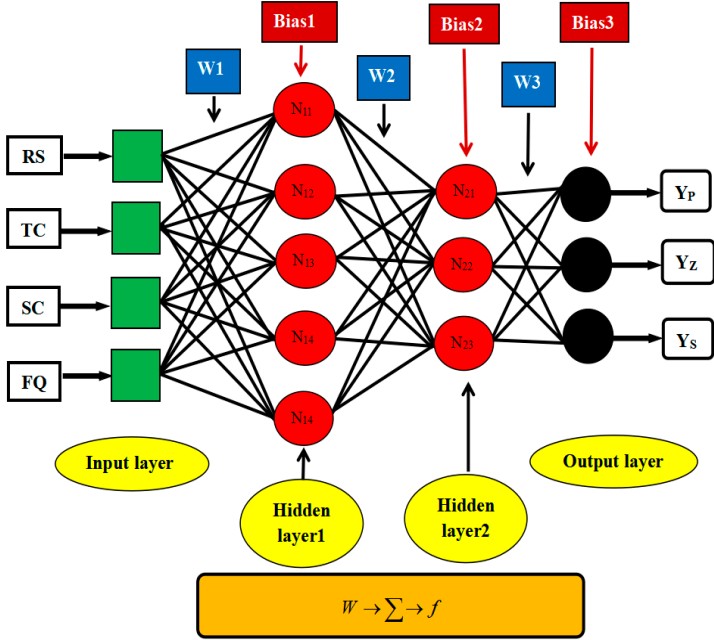

**Figure 6.** The artificial neural network architecture of threshing performance.

### 2.3.5. Weight Optimization

"Training" or "learning" is the process of optimizing the connection weights. The goal is to identify a global solution to what is typically a highly nonlinear optimization problem. The method most commonly used for finding the optimum weight combination for feedforward neural networks is the backpropagation algorithm [37], which is based on first-order gradient descent. Feedforward networks trained with the backpropagation algorithm have been applied successfully to numerous agricultural engineering problems [38,39], hence, also used in this work.

### 2.3.6. Stopping Criteria and ANN Model Validation

The following are conditions for the neural network to stop: 1. Meet the accuracy requirements; 2. Complete the maximum number of iterations.

Backpropagation works by minimizing a cost function. The mean squared error (*MSE*) is the most common cost function.

Validation data were used to validate the performance of the trained model once the training phase of the model was completed. Additionally, the validation set was used to determine the optimum number of hidden layer nodes and the optimum internal parameters (learning rate, momentum, and initial weights). The *MSE* was used to validate the performance of the ANN in terms of the different number of hidden layer nodes according to Equation (4).

$$MSE = \frac{\sum\limits_{i=1}^{m} (y_i - \hat{y}_i)^2}{m} \tag{4}$$

The evaluation parameters metrics of root mean square error (*RMSE*) [40], correlation coefficient (*R*), and mean absolute error (*MAE*) were utilized to assess the performance of the models by comparing the target and output values of networks.

$$RMSE = \sqrt{\frac{\sum\limits_{i=1}^{m} (y_i - \hat{y}_i)^2}{m}} \tag{5}$$

$$R = \sqrt{\frac{\left(\sum\limits_{i=1}^{m} (y_i - \overline{y})(\hat{y}_i - \overline{\hat{y}})\right)^2}{\sum\limits_{i=1}^{m} (y_i - \overline{y})^2 \bullet \sum\limits_{i=1}^{m} (\hat{y}_i - \overline{\hat{y}})^2}} \tag{6}$$

$$MAE = \frac{1}{m}\sum\limits_{i=1}^{m} |y_i - \hat{y}_i| \tag{7}$$

The *RMSE*, *R*, and *MAE* values were calculated in all stages: training; validating; and testing. Where $y_i$, $\hat{y}_i$ are the observed value and predicted values, $\overline{y}$, $\overline{\hat{y}}$ are the average observed and predicted values, and *m* is the total number of points in each dataset, respectively. Using this parameter aids in selecting the best structure and network and provides the possibility of understanding the proximity of the model.

After model construction, the variable parameters of the experimental trials were entered as the new input model, and the actual results were compared with the model. Microsoft Excel 2016 software was used to analyze the correlation coefficient between the actual results and the output of the neural network model.

## 3. Results

### 3.1. Evaluation of the Number of Hidden Layer Nodes

The BP network has a varied number of nodes in the hidden layer, and the hidden nodes affect the error of the output connected neurons [41]. If the number of neurons

in the hidden layer is too small, the network's ability to learn is limited, resulting in the need for more training to decrease its fault tolerance. On the other hand, network iterations will increase with too many neurons, thereby extending the training time of the network, and reducing the generalization capacity of the network, resulting in a decrease in predictive ability. The optimal number of nodes needs to be explored to confirm the effect of different nodes on network performance. In practical situations, the number of nodes in the hidden layer is selected by first determining the approximate range of the number of nodes using the empirical formula before using the step-wise test strategy to establish the best number of nodes with the smallest error by training and comparing the networks with different neurons. The best number of hidden layer nodes can be derived from the following formula [42,43]:

$$l = \sqrt{(m+n)} + a \tag{8}$$

where $l$ represents the number of neurons in the hidden layer, $n$ denotes the number of neurons in the input layer, $m$ is the number of neurons in the output layer, $a$ is the constant, and $1 < a < 10$. According to this formula, the value range of the hidden layer nodes of the network was 4–12, and the performance of the artificial neural network under different numbers of nodes is shown in Figure 7. When the number of hidden layers was 5, the minimum *MSE* was 0.00080796, indicating superior model performance.

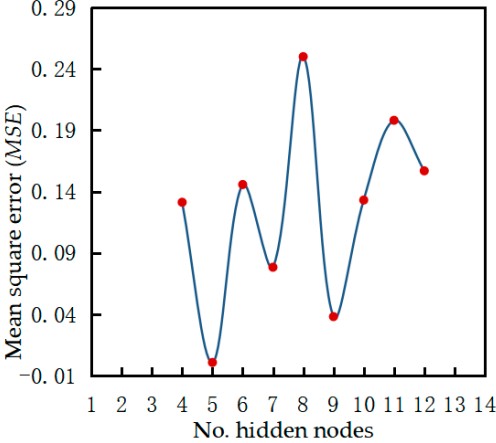

**Figure 7.** Performance of artificial neural network models with different hidden layer nodes (learning rate = 0.1 and training goal = 0.001).

Table 5 summarizes the predictive performance of the optimal neural network. The findings showed a validation set of $R = 0.979$, *RMSE* of 0.138, and *MAE* of 0.153. The ANN model with a 4-5-3-3 structure performed effectively. Table 5 further shows the results of the model, which were generally consistent with those obtained during training and testing, indicating that the model can generalize within the range of data used for training.

**Table 5.** Artificial Neural Network Results.

| Dataset | *R* | *RMSE* | *MAE* |
|---|---|---|---|
| Training set | 0.97596 | 0.079148 | 0.14100 |
| Validation set | 0.97981 | 0.13823 | 0.15260 |
| Testing set | 0.99041 | 0.086466 | 0.13543 |

Based on the data shown in Figure 8, the error curves of the model training sample, the corrected sample, and the test sample were well correlated. The curve trend slowly decreased, indicating that the network was trained on the training data. To avoid overfitting with the validation data, the *MSE* between the initial fitting and validation will become

smaller and smaller, but as the network begins to overfit the training data, the *MSE* will become larger. In the default setting, the training ends when the validation error is added six consecutive times, and the best performance is obtained from the lowest validation error period (drawing circle). Finally, the obtained best artificial neural network parameters are shown in Table 6. Figure 9 shows the training state of the model training phase.

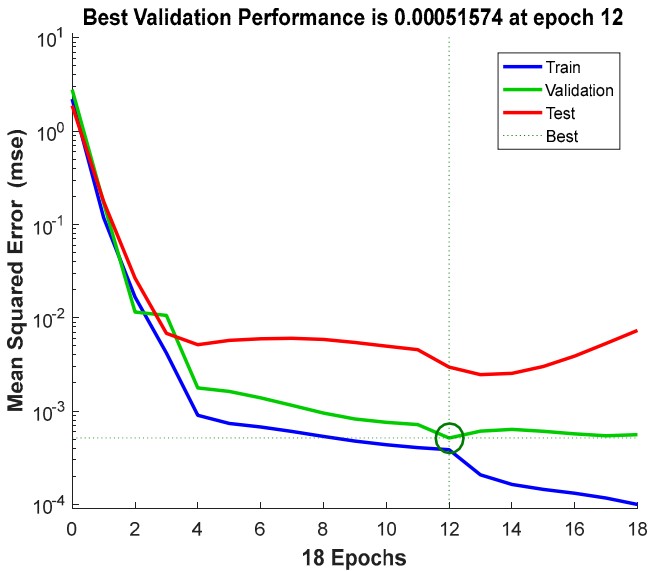

**Figure 8.** The neural network training performance (epoch 18, validation stop).

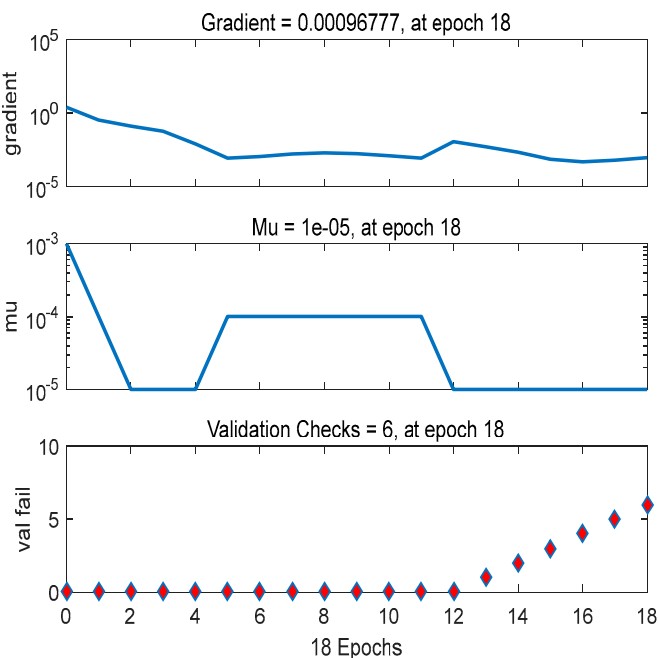

**Figure 9.** The neural network training state (epoch 18, validation stop).

**Table 6.** Optimum ANN parameters for the design of the model.

| Sr.No. | Parameter | Description |
|---|---|---|
| 1 | No. of input nodes | Varying from 1 to 25 in the cascaded training procedure |
| 2 | No. of output nodes | 3 |
| 3 | No. of hidden layers | 2 |
| 4 | No. of neurons in the hidden layer (Hn) | 5–3 |
| 5 | Training rule | Levenberg-Marquardt (LM) |
| 6 | Activation function | Sigmoid |
| 7 | Network type | Feed-forward (FF) |
| 8 | Training method | Backpropagation algorithm |

### 3.2. Evaluation of Prediction Results

The regression curves for assessing the accuracy of the ANN estimation are shown in Figure 10. Estimates of the threshing performance of the ANN were evaluated by regression analysis between the predicted and experimental data. To validate the ANN model, we applied the estimation and regression methods. The regression value for the threshing characteristics was calculated as 0.9525. Figure 10 displays the optimal curve resulting from multiple iterations of the $R^2$ curve.

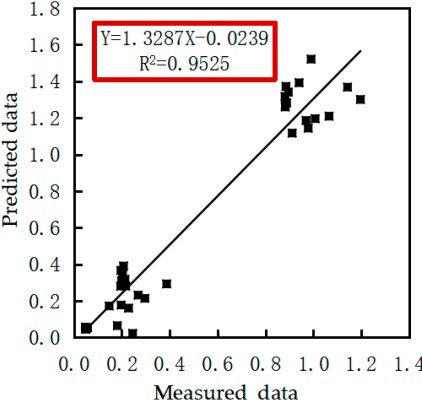

**Figure 10.** The predicted and the measured values of threshing performance.

### 3.3. Sensitivity Analysis

Sensitivity analysis was performed to examine the sensitivity of the various factors influencing the threshing characteristics, and Table 7 shows the effects of the various input factors. As shown, different input variable values, i.e., the size of different sensitivity, reflected the effect of the input variable on the output variable. RS affected the predicted threshing performance when the network values had distinct input variables. However, the relative importance of the remaining input variables varied based on changes in the input variables. RS was the most important input in all trials followed by TC, SC, and FQ. Sensitivity analysis revealed that RS, TC, and SC were the most vital factors affecting threshing performance, with an average relative importance of 15.00%, 14.89%, and 14.32%, respectively. The results further showed that FQ had a minimal effect on threshing performance, with an average relative importance of 11.65%.

**Table 7.** Sensitivity Analyses of the Relative Importance of Artificial Neural Network Input Variables.

| Trial No. | Relative Importance for Input Variables | | | |
| :---: | :---: | :---: | :---: | :---: |
| | RS | TC | SC | FQ |
| 1 | 0.2417 | 0.1083 | 0.1208 | 0.1013 |
| 2 | 0.1831 | 0.1575 | 0.1468 | 0.086 |
| 3 | 0.1426 | 0.1278 | 0.1892 | 0.0828 |
| 4 | 0.0507 | 0.2191 | 0.1274 | 0.1805 |
| Average | 0.1500 | 0.1489 | 0.1432 | 0.1165 |

## 4. Discussion

Threshing is one of the most critical operations of combine harvesters during grain production, which is a complex, nonlinear, multi-parameter physical process. The working performance index of the threshing device has a significant on the separation, cleaning, and other parts and the working quality of the whole machine and has always been one of the main concerns of the engineered design. A flexible threshing device has the advantage of reducing the crushing rate of rice grain. Therefore, a comprehensive and accurate design of a flexible threshing performance evaluation model has important theoretical value and practical significance. In this study, the BP artificial neural network was used to model the threshing performance factors based on four factors: RS, TC, SC, and FQ. Determining the optimal network architecture is related to the number of hidden layers and neurons. The optimum network geometry was found to be 4-5-3-3 by evaluating different number of hidden layer nodes in this study. The performance of the ANN model was verified by comparing the predicted dataset with the experimental results (measured data). The sensitivity analysis performed for the described ANN model indicated that four working variables of the flexible threshing device had the greatest contribution to threshing performance attributes compared to FQ. These results can guide the optimal design of a flexible threshing cylinder to achieve the maximum performance of the device.

## 5. Conclusions

This study analyzed different numbers of hidden layer nodes and found that when the number of hidden layer nodes was five, the minimum *MSE* was 0.00080796, indicating that the model performed well. The results indicated that backpropagation neural networks could predict the threshing performance of the flexible threshing device with an acceptable degree of accuracy ($R = 0.980$, $RMSE = 0.138$, $MAE = 0.153$). The built neural network model prediction predicted the performance of the flexible threshing device well. The regression determination coefficient $R^2$ between the predicted and experimental data was 0.953, indicating that the predicted data of the built neural network model was in good agreement with the experimental data. The ANN method is an effective method for predicting the threshing performance of flexible threshing devices in rice. The established artificial neural network model exhibited stable prediction of the threshing performance of the flexible threshing device during operation. The sensitivity analysis revealed that RS, TC, and SC are important factors affecting the performance of the flexible threshing device, with an average relative importance of 15.00%, 14.89%, and 14.32%, respectively. FQ had the least impact on threshing performance, with an average threshing relative importance of 11.65%. These results can guide the optimal design of flexible threshing cylinders and improve the performance of the flexible threshing device.

**Author Contributions:** Conceptualization, L.M. and F.X.; methodology, L.M.; software, L.M.; validation, L.M., F.X. and D.L.; formal analysis, L.M.; investigation, D.L.; resources, X.W. and Z.Z.; data curation, X.W. and D.L.; writing—original draft preparation, L.M.; writing—review and editing, L.M., F.X. and Z.Z.; visualization, L.M.; supervision, F.X.; project administration, D.L.; funding acquisition, F.X. All authors have read and agreed to the published version of the manuscript.

**Funding:** This research was funded by Hunan High-Tech Industry Technology Leading Plan Project (Science and Technology Research category) (2020NK2002); Hunan Agricultural Machinery Equipment and Technology Innovation Research and Development Project (Xiangcai Agricultural Index (2021) No.47); and Hunan Agricultural Machinery Equipment and Technology Innovation Research and Development Project (Xiangcai Agricultural Index [2020] No.107).

**Institutional Review Board Statement:** Not applicable.

**Informed Consent Statement:** Not applicable.

**Data Availability Statement:** The data presented in this study are available on request from the corresponding author.

**Conflicts of Interest:** The authors declare no conflict of interest.

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
