# Peer review of "An Application of Artificial Neural Network for Predicting Threshing Performance in a Flexible Threshing Device"

_agriculture, doi:10.3390/agriculture13040788_

Round 1

Reviewer 1 Report

It’s my honor to review the manuscript “An Application of Artificial Neural Network for Predicting Threshing Performance in Flexible Threshing Device” (agriculture-2322915).

The manuscript was that explored factors influencing the performance of a flexible threshing device and the nonlinear relationship between the threshing performance index and obtained the ANN prediction model of flexible threshing performance index, the results showed that the ANN method is an effective approaches in predicting the threshing performance of the flexible threshing device in rice, the model will replace the repeatability, time-consuming and expensive bench experiment in the flexible threshing system. which provide the basis for the optimization of the flexible threshing device.

With a clear structure and rigorous logic, the manuscript strengthened itself by a scientific test method, whose result would be significant to the further research on the optimization of the flexible threshing device. However, the following problems is still need to be revised or explained.

1. Figure 2 is about The artificial neural network algorithm flow chartbut the word of “prediction” of figure in line 199 is wrong in spelling.

2. There is a advice, in line 219-227, the sensitivity analysis is introduced, if it is explained by  figure or formulais it more clear?

3. Figure 1 is Flexible threshing experiment device, the arrow in the figure is so big, please use the arrow matching the figure size

4. The statement in the text needs further retouching, which has increased the consistency of the statement.

Reviewer 2 Report

First, the flexible threshing is a complex, nonlinear, multi-parameter physical process. we explored factors influencing the performance of a flexible threshing device and the nonlinear relationship between the threshing performance index and obtained the ANN prediction model of flexible threshing performance index, the results showed that the ANN method is an effective approaches in predicting the threshing performance of the flexible threshing device in rice, the model will replace the repeatability, time-consuming and expensive experiment in the flexible threshing system. which provide the basis for the optimization of the flexible threshing device.

 Second, the sensitivity analysis performed based on the described ANN model indicated that four working variables (the rotated speed of cylinder (RS), the threshing clearance of concave sieve(TC), and the separation clearance of concave sieve (SC))of the flexible threshing device had the greatest contribution for threshing performance attributes compared to feeding quantity (FQ). These results can guide the optimal design of flexible threshing device to achieving the maximum performance of the device.

 Thirdly, we evaluate the number of hidden layer nodes, When the number of hidden layers is 5, the minimum MSE is 0.00080796, indicating superior model performance. And summarized the predictive performance of the optimal neural network. The findings showed a validation set of R = 0.980, RMSE of 0.138, and MSE of 0.153. The ANN model with a 4-5-3-3 structure performed effectively. These work can provide a method for us to find the optimal neural network structure.

Author Response

Reply: We are very grateful to the referee for his  positive comments.
